# Approaches to the Synthesis of Dicarboxylic Derivatives of Bis(pyrazol-1-yl)alkanes

**DOI:** 10.3390/molecules26020413

**Published:** 2021-01-14

**Authors:** Nikita P. Burlutskiy, Andrei S. Potapov

**Affiliations:** 1Kizhner Research Center, National Research Tomsk Polytechnic University, 30 Lenin Ave., 634050 Tomsk, Russia; npb1@tpu.ru; 2Nikolaev Institute of Inorganic Chemistry, Siberian Branch of the Russian Academy of Sciences, 3 Lavrentiev Ave., 630090 Novosibirsk, Russia

**Keywords:** pyrazole, bis(pyrazol-1-yl)alkanes, carboxylation, oxalyl chloride, dicarboxylic acids, alkylation, superbasic medium

## Abstract

Carboxylation of bis(pyrazol-1-yl)alkanes by oxalyl chloride was studied. It was found that 4,4′-dicarboxylic derivatives of substrates with electron-donating methyl groups and short linkers (from one to three methylene groups) can be prepared using this method. Longer linkers lead to significantly lower product yields, which is probably due to instability of the intermediate acid chlorides that are initially formed in the reaction with oxalyl chloride. Thus, bis(pyrazol-1-yl)methane gave only monocarboxylic derivative even with a large excess of oxalyl chloride and prolonged reaction duration. An alternative approach involves the reaction of ethyl 4-pyrazolecarboxylates with dibromoalkanes in a superbasic medium (potassium hydroxide–dimethyl sulfoxide) and is suitable for the preparation of bis(4-carboxypyrazol-1-yl)alkanes with both short and long linkers independent of substitution in positions 3 and 5 of pyrazole rings. The obtained dicarboxylic acids are interesting as potential building blocks for metal-organic frameworks.

## 1. Introduction

Bis(pyrazol-1-yl)alkanes are bidentate ligands widely used for the synthesis of the coordination compounds [1], some of which were shown to exhibit catalytic [2,3,4,5,6], anticancer [7,8], antibacterial [9] and SOD-like activity [10,11], and electroluminescent properties [12,13]. The properties of the ligands can be varied by introducing the functional groups into the pyrazole rings [14,15,16,17], besides, carboxylic groups themselves can act as donor groups for the formation of coordination bond with the metal ions. Carboxy-substituted pyrazoles were used for the construction of highly porous metal-organic frameworks [18,19,20,21]. Carboxylic acids based on bis(pyrazol-1-yl)alkanes are much less explored and only a few research papers were published so far [22,23,24,25]. In addition, a series of works devoted to the synthesis of metal-organic frameworks based on structurally related bis(4-carboxyphenylpyrazol-1-yl)methane was carried out by Sumby et al., the presence of the free chelating units in the structure of the framework allowed them to prepare catalysts with single metal sites [26,27].

In this contribution we report a facile synthesis of a series of 4,4′-dicarboxy-substituted bis(pyrazol-1-yl)alkanes with varied linker length and substitution in positions 3 and 5 of the pyrazole rings.

## 2. Results and Discussion

To introduce carboxyl groups into bis(pyrazol-1-yl)alkanes, a reaction with oxalyl chloride was used, in which it was both a reagent and a solvent. Previously oxalyl chloride was successfully used for the carboxylation of 1-phenyl- and 1-alkylpyrazoles [28]. In this reaction, a pyrazole-containing derivative of oxalic acid chloride is initially formed, which is converted to a carboxylic acid chloride with the release of carbon monoxide. The acid chloride is hydrolyzed without isolation to form a carboxylic acid. Other methods for the introduction of carboxylic groups into pyrazole rings include oxidation of alkyl [29,30] or formyl [31] groups, substitution of halogens via intermediate organolithium derivatives [32], and hydrolysis of trichloromethyl derivatives [33].

Carboxylation of bis(3,5-dimethylpyrazol-1-yl)methane **1a** with subsequent hydrolysis gave the corresponding dicarboxylic acid **1b** in high yield (Scheme 1). Similarly, 1,2-bis(3,5-dimethylpyrazol-1-yl)ethane **2a** and 1,3-bis(3,5-dimethylpyrazol-1-yl)propane **3a** gave dicarboxylic acids **2b** and **3b**, albeit in lower yield (Scheme 1). When bis(3,5-dimethylpyrazol-1-yl)alkanes with longer spacers (tetramethylene, (CH_2_)_4_ and longer) were introduced into the reaction, full conversion of the starting materials was achieved (GC–MS control), but no dicarboxylic acids were isolated. Instead, mixtures of monosubstituted carboxy-derivatives (not isolated) and starting substrates were obtained, which is probably a result of instability of the intermediate acid chlorides leading to their decarboxylation during the hydrolysis.

To overcome the instability of acid chlorides their methanolysis instead of hydrolysis was attempted and indeed led to the dimethyl esters **4c**–**6c**, although in low yields (Scheme 2). Alkaline hydrolysis of the esters proceeded smoothly with the formation of dicarboxylic acids **4b**–**6b** (Scheme 2).

Carboxylation by oxalyl chloride proved to be poorly suitable for bis(pyrazol-1-yl)alkanes without electron-donating methyl groups. Thus, only in the case of 1,3-bis(pyrazol-1-yl)propane **3d** a dicarboxylation product could be isolated, while bis(pyrazol-1-yl)methane **1d** gave solely a monocarboxylated product **7** (Scheme 3). Even a large excess of oxalyl chloride and prolonged heating did not lead to the formation of the dicarboxylic acid. For derivatives with longer spacers (four to six methylene groups) only traces of carboxylation products (after conversion to methyl esters) were detected (for substrate **4c**) but were not isolated (Scheme 3).

In order to gain insight into the reasons for low reactivity of bis(pyrazol-1-yl)methane **1d**, DFT calculations of charge distribution in the starting compound **1d** and compounds **1a**, **3a**, and **3d** for comparison and the corresponding monosubstituted acid chlorides formed during the first electrophilic substitution step were carried out. The structures were optimized at the B3LYP 6-31G(d) level of theory and absence of imaginary frequencies confirmed that the found stationary points correspond to minima on the potential energy surface. In order to get a more precise electron density distribution, MP2 6-31G(d) single point calculations were carried out for the found structures. Next, charges on carbon atoms (*q*_C_) in position 4 of the pyrazole ring (since an electrophilic attack is directed at this position) and the hydrogen atom bound to it (*q*_H_,) were calculated using the Bader’s theory of atoms in molecules [34]. The calculation results are presented in Table 1.

The following conclusions can be drawn from the obtained charge distribution:

(1)The introduction of methyl groups into pyrazole rings (in pairs of compounds **1a**–**1d** and **3a**–**3d**) noticeably increases the negative charge at position 4 of the heterocycle, i.e., makes it more active in the electrophilic substitution reaction, and the effect of electron-donor groups is best manifested when comparing the sum of charges on carbon and hydrogen atoms;(2)An increase in the length of the linker from one to three methylene groups also increases the excess negative charge at position 4, which is apparently associated with the negative inductive effect of the pyrazole ring;(3)The introduction of an electron-withdrawing acid chloride group into one of the pyrazole rings deactivates the other cycle in the electrophilic substitution reaction, and to a greater extent, deactivation manifests itself in pyrazole derivatives without methyl substituents, and with a short methylene linker.

Based on the charge distribution, bis(pyrazol-1-yl)methane **1d** is the least active in electrophilic substitution reactions, the charge on the 4-CH group in which is close to the charge on this group in the acid chloride of the propane derivative **3d′**. Experimental data show that monochloro anhydride **3d′** can undergo carboxylation with the formation of dicarboxylic acid **3e**. At the same time, bis(pyrazol-1-yl)methane **1d** at the first step of electrophilic substitution gives acid chloride **1d′**, in which position 4 of the other pyrazole ring is so deactivated by the electron-withdrawing effect of the already substituted heterocycle that the reaction halts on monochloroanhydride **1d′**, the hydrolysis of which gives monocarboxylic acid **7**. Therefore, the pyrazole ring with an electron-withdrawing functional group (i.e., chlorocarbonyl) is a substituent with a strong negative inductive effect, which affects the reactivity of the neighboring pyrazole ring located even across two aliphatic bonds.

Taking into account the limitations of direct carboxylation of bis(pyrazol-1-yl)alkanes, an alternative approach involving the double alkylation of 4-pyrazolecarboxylic acid and its 3,5-dimethylderivative by α,ω-dibromoalkanes was evaluated. Ethyl esters of 4-pyrazolecarboxylic acid and 3,5-dimethyl-4-pyrazole carboxylates were smoothly alkylated by α,ω-dibromoalkanes in a superbasic KOH-DMSO system and gave the corresponding diethyl dicarboxylates in high yields (Scheme 4). Alkaline hydrolysis of the diesters and subsequent neutralization gave the target dicarboxylic acids, including the ones with longer spacers, unavailable by direct carboxylation (Scheme 4). Unfortunately, derivatives of 1,2-bis(pyrazol-1-yl)ethane cannot be prepared by this route, since, as it is known, the reaction of pyrazole and its derivatives with 1,2-dibromoethane leads to the formation of 1-vinylpyrazoles, which undergo hydrolysis to the starting materials upon isolation [35].

Since both alkylation and ester hydrolysis take place under basic catalysis, we explored the possibility to carry out the synthesis of diacid **1e** in a one-pot process. Indeed, after the alkylation was complete (TLC control), two additional equivalents of KOH and excess of water were added to the reaction mixture, which gave the hydrolysis product in several minutes in 92% overall yield.

## 3. Materials and Methods

Gas chromatography-mass spectrometry analysis was performed using Agilent 7890A gas chromatograph (Santa Clara, CA, USA) equipped with Agilent MSD 5975C mass-selective detector with quadrupole mass-analyzer (electron impact ionization energy 70 eV). NMR spectra were recorded on Bruker DRX400 and Bruker Advance 500 instruments (Billerica, MA, USA), solvent residual peaks were used as internal standards. Elemental analyses were carried out on a Vario Micro-Cube analyzer (Elementar Analysensysteme GmbH, Langenselbold, Germany). IR-spectra of solid samples were recorded on Agilent Cary 630 FT IR (Santa Clara, CA, USA) spectrophotometer equipped with diamond ATR accessory. Melting points of compounds **1b**, **3b**, and **1d** (with melting points higher than 300 °C) were determined in helium atmosphere on NETZSCH TG 209 F1 thermoanalyzer (NETZSCH TAURUS Instruments GmbH, Weimar, Germany) with the heating rate of 10°/min.

The calculations were performed using the Gaussian 09 package, revision D.01 [36]. Atomic charges were calculated using the AIMAll Professional 10.05.04 package.

Ethyl 4-pyrazolecarboxylate [37], ethyl 3,5-dimethyl-4-pyrazolecarboxylate [25], bis(pyrazol-1-yl)alkanes **1a** and **1d** [38], **2a** [35], **3d** and **3a** [39], **4a**–**6a,** and **4d**–**6d** [40] were prepared following the literature procedures. Dimethyl sulfoxide was distilled from KOH under vacuum (10 Torr) prior to use. Other commercially reagents and solvents were of reagent grade and used as received. FT-IR and NMR plots of the synthesized compounds can be found in the Appendix A.

*Bis(3,5-dimethylpyrazol-1-yl)methane-4,4′-dicarboxylic acid* (**1b**). Method A (carboxylation by oxalyl chloride). Oxalyl chloride (1.91 g, 1.29 mL, 15 mmol) was added dropwise to 0.51 g (2.5 mmol) of compound **1a**. The mixture was refluxed during 3 h, after which the excess of oxalyl chloride was removed in vacuum of water aspiration pump. The residue was treated by 10 mL of water, the precipitate was filtered and washed with water. Yield 0.606 g, 83%, colorless power, m.p. 322–324 °C (EtOH, dec.). C_13_H_16_N_4_O_4_ (292.12): calcd. C 53.42; H 5.52; N 19.17; found C 53.4; H 5.7; N 19.0. ^1^H-NMR (400 MHz, D_2_O): δ 2.21 (s, 6H, 3-CH_3_), 2.51 (s, 6H, 5-CH_3_), 6.16 (s, 2H, CH_2_), ppm. ^13^C-NMR (100 MHz, D_2_O): δ 10.1, 12.2, 58.6, 113.9, 148.0, 152.6, 189.5, ppm. FT-IR (cm^−1^): ν = 2598 (w), 2532 (w), 1668 (m), 1554 (m), 1472 (w), 1435 (m), 1370 (w), 1316 (m), 1251 (m), 1164 (m), 1119 (m), 1040 (w), 1005 (w), 928 (w), 868 (w), 790 (m), 770 (m), 732 (w), 691 (m). Compounds **2b** and **3b** can be prepared similarly.

Method B (hydrolysis of diethyl diesters). Compound **1f** (0.696 g, 2 mmol) was suspended in 10 mL of the aqueous solution of KOH (0.448 g, 8 mmol) and heated at 80 °C until complete dissolution of a starting diester. After cooling, the obtained solution of potassium salts was acidified by concentrated HCl solution, the resulting precipitate was filtered, washed with water and dried. Yield 0.514 g, 88%.

*1,2-Bis(3,5-dimethylpyrazol-1-yl)ethane-4,4′-dicarboxylic acid* (**2b**) was prepared similarly to compound **1b** by Method A. Yield 38%, colorless crystals, m.p. 224–225 °C (H_2_O). C_14_H_18_N_4_O_4_ (306.32): calcd. C 54.89; H 5.92; N 18.29; found C 54.7; H 6.0; N 18.1. ^1^H-NMR (400 MHz, (CD_3_)_2_SO): δ 2.09 (s, 6 H, 3-CH_3_), 2.25 (s, 6 H, 5-CH_3_), 4.43 (s, 4 H, CH_2_), ppm. ^13^C-NMR (100 MHz, (CD_3_)_2_SO): δ 9.6 (5-CH_3_), 13.1 (3-CH_3_), 47.4 (CH_2_), 113.1 (C^4^(Pz)), 145.8 (C^5^(Pz)), 149.9 (C^3^(Pz)), 184.0 (COOH), ppm. FT-IR (cm^−1^): 3500 (br., COOH), 1664 (C=O), 1535 (Pz), 1007 (Pz breathing).

*1,3-Bis(3,5-dimethylpyrazol-1-yl)propane-4,4′-dicarboxylic acid* (**3b**) was prepared similarly to compound **1b** by Methods A and B. Yield 39% (Method A) and 84% (Method B), colorless crystals, m.p. 290–292 °C (EtOH). C_15_H_20_N_4_O_4_ (320.35): calcd. C 56.24; H 6.29; N 17.49; found C 56.4, H 6.1, N 17.1. ^1^H-NMR (400 MHz, (CD_3_)_2_SO): δ 2.07 (p, 2 H, β-CH_2_), 2.24 (s, 6 H, 3-CH_3_), 2.42 (s, 6 H, 5-CH_3_), 4.07 (t, 4 H, CH_2_), ppm ^13^C-NMR (100 MHz, (CD_3_)_2_SO): δ 10.4 (5-CH_3_), 13.2 (3-CH_3_), 28.5 (PzCH_2_CH_2_), 45.3 (PzCH_2_CH_2_), 112.9 (C^4^(Pz)), 144.9 (C^5^(Pz)), 149.3 (C^3^(Pz)), 185.0 (COOH) ppm. FT-IR (cm^−1^): 3430 (br., COOH), 1651 (C=O), 1536 (Pz), 1005 (Pz breathing).

*1,4-Bis(3,5-dimethylpyrazol-1-yl)butane-4,4′-dicarboxylic acid* (**4b**) was prepared similarly to compound **1b** by Method B from diesters **4c** or **4g**. Yield 80%, colorless crystals, m.p. 273–275 °C (reprecipitation; dec.). C_16_H_22_N_4_O_4_ (334.38): calcd. C 57.47; H 6.63; N 16.76; found C 57.1; H 6.5; N 16.9. ^1^H-NMR (400 MHz, (CD_3_)_2_SO): δ 1.66 (p, 4H, β-CH_2_), 2.25 (s, 6H, 3-CH_3_), 2.43 (s, 6H, 5-CH_3_), 3.98 (t, 4H, α-CH_2_), 12.12 (s, 2 H, COOH), ppm. ^13^C-NMR (100 MHz, (CD_3_)_2_SO): δ 11.1, 14.5, 26.8, 47.9, 109.4, 143.7, 149.4, 165.8 ppm. FT-IR (cm^−1^): ν = 3464 (w), 2932 (w), 2649 (w), 1674 (m), 1556 (w), 1506 (w), 1458 (w), 1431 (m), 1394 (w), 1366 (w), 1302 (m), 1264 (m), 1189 (m), 1118 (m), 1089 (m), 1038 (w), 1005 (w), 887 (w), 863 (w), 787 (m), 757 (s).

*1,5-Bis(3,5-dimethylpyrazol-1-yl)pentane-4,4′-dicarboxylic acid* (**5b**) was prepared similarly to compound **1b** by Method B from diesters **5c** or **5g**. Yield 70%, colorless crystals, m.p. 243–244 °C (reprecipitation; dec.). C_17_H_24_N_4_O_4_ (348.40): calcd. C 58.61; H 6.94; N 16.08; found C 58.3; H 6.7; N 15.9. ^1^H-NMR (400 MHz, (CD_3_)_2_SO): δ 1.20 (p, 2 H, γ-CH_2_), 1.69 (p, 4 H, β-CH_2_), 2.25 (s, 6 H, 3-CH_3_), 2.42 (s, 6 H, 5-CH_3_), 3.95 (t, 4 H, α-CH_2_), ppm. ^13^C-NMR (100 MHz, (CD_3_)_2_SO): δ 11.1, 14.5, 23.6, 29.3, 48.3, 109.4, 143.6, 149.3, 165.8, ppm. FT-IR (cm^−1^): ν = 2933 (w), 2866 (w), 1689 (m), 1669 (m), 1545 (m), 1499 (w), 1435 (w), 1384 (w), 1301 (w), 1268 (m), 1237 (w), 1216 (m), 1165 (m), 1114 (m), 1043 (w), 1012 (w), 824 (w), 787 (w), 752 (m).

*1,6-Bis(3,5-dimethylpyrazol-1-yl)hexane-4,4′-dicarboxylic acid* (**6b**) was prepared similarly to compound **1b** by Method B from diesters **6c** or **6g**. Yield 67%, colorless crystals, m.p. 193–195 °C (reprecipitation; dec.). ^1^H-NMR (400 MHz, (CD_3_)_2_SO): δ 1.23 (p, 4 H, γ-CH_2_), 1.66 (p, 4 H, β-CH_2_), 2.26 (s, 6 H, 3-CH_3_), 2.42 (s, 6 H, 5-CH_3_), 3.94 (t, 4 H, α-CH_2_), 12.13 (s, 2 H, COOH), ppm. ^13^C-NMR (100 MHz, (CD_3_)_2_SO): δ 11.1, 14.5, 26.0, 29.6, 48.3, 109.3, 143.60, 149.3, 165.8, ppm. FT-IR (cm^−1^): ν = 2933 (w), 2861 (w), 1697 (s), 1543 (m), 1497 (m), 1470 (m), 1430 (m), 1380 (m), 1277 (m), 1235 (m), 1178 (m), 1108 (s), 1044 (w), 1002 (m), 977 (w), 862 (m), 787 (m), 746 (s), 730 (m), 664 (w).

*1,4-Bis(4-methoxycarbonyl-3,5-dimethylpyrazol-1-yl)butane* (**4c**). Oxalyl chloride (1.91 g, 1.29 mL, 15 mmol) was added dropwise to 0.51 g (2.5 mmol) of compound **4a**. The mixture was heated in a screw-cup vial for 33 h, after which the reaction mixture was transferred to 50 mL of chloroform and the solvent was evaporated in vacuum of water aspiration pump to remove the excess of oxalyl chloride. Methanol (5 mL) was added dropwise to the residue, 50 mL of water was added and the precipitate was filtered and washed with water. Yield 27%, colorless crystals, m.p. 129–132 °C (EtOAc/hexane, 1:1). C_18_H_26_N_4_O_4_ (362.42): calcd. C 59.65; H 7.23; N 15.46; found C 59.8; H 7.5; N 15.2. ^1^H-NMR (400 MHz, CDCl_3_): δ =1.80 (p, 4 H, β-CH_2_), 2.40 (s, 6 H, 3-CH_3_), 2.47 (s, 6 H, 5-CH_3_), 3.82 (s, 6 H, OCH_3_), 4.01 (t, 4 H, α-CH_2_), ppm. ^13^C-NMR (100 MHz, CDCl_3_): δ 11.2, 14.2, 26.9, 48.1, 50.9, 109.4, 143.7, 150.4, 164.9, ppm. FT-IR (cm^−1^): ν = 2988 (w), 2930 (w), 2861 (w), 1685 (s), 1544 (s), 1493 (m), 1442 (s), 1430 (s), 1303 (m), 1261 (m), 1223 (m), 1191(m), 1175 (m), 1104 (s), 1082 (s), 1041 (m), 1005 (m), 960 (m), 842 (w), 803 (m), 785 (s), 768 (m), 657 (w). *m*/*z*: 362 (35%), 347 (12%), 331 (16%), 315 (4%), 303 (4%), 209 (29%), 195 (31%), 181 (57%), 167 (100%).

*1,5-Bis(4-methoxycarbonyl-3,5-dimethylpyrazol-1-yl)pentane* (**5c**) was prepared similarly to compound **4c**. Yield 34%, colorless crystals, m.p. 86–87 °C (EtOAc/hexane, 1:1). C_19_H_28_N_4_O_4_ (376.46): calcd. C 60.62; H 7.50; N 14.88; found C 60.3; H 7.4; N 14.6. ^1^H-NMR (400 MHz, CDCl_3_): δ 1.30 (p, 2 H, γ-CH_2_), 1.81 (p, 4 H, β-CH_2_), 2.40 (s, 6 H, 3-CH_3_), 2.48 (s, 6 H, 5-CH_3_), 3.82 (s, 6 H, OCH_3_), 3.99 (t, 4 H, α-CH_2_), ppm. ^13^C-NMR (100 MHz, CDCl_3_): δ 11.2, 14.2, 23.7, 29.5, 48.5, 50.9, 109.3, 143.6, 150.3, 164.9, ppm. FT-IR (cm^−1^): ν = 2934 (w), 2863 (w), 1690 (s), 1546 (s), 1495 (w), 1439 (s), 1377 (w), 1324 (w), 1297 (s), 1271 (m), 1236 (m), 1213 (s), 1192 (m), 1160 (m), 1102 (s), 1040 (w), 1001 (w), 959 (m), 856 (w), 825 (w), 805 (w), 785 (s), 736 (w), 713 (w). *m*/*z*: 376 (27%), 361 (11%), 345 (14%), 329 (5%), 317 (4%), 223 (24%), 209 (39%), 195 (25%), 181 (37%), 167 (100%).

*1,6-Bis(4-methoxycarbonyl-3,5-dimethylpyrazol-1-yl)hexane* (**6c**) was prepared similarly to compound **4c**. Yield 30%, colorless crystals, m.p. 112–114 °C (EtOAc/hexane, 1:1). ^1^H-NMR (400 MHz, CDCl_3_): δ 1.23 (p, 2 H, γ-CH_2_), 1.65 (p, 4 H, β-CH_2_), 2.26 (s, 6 H, 3-CH_3_), 2.43 (s, 6 H, 5-CH_3_), 3.72 (s, 6 H, OCH_3_), 3.95 (t, 4 H, α-CH_2_), ppm. ^13^C-NMR (100 MHz, CDCl_3_): δ 11.2, 14.5, 25.9, 29.5, 48.4, 51.1, 108.6, 143.7, 149.1, 164.6, ppm. FT-IR (cm^−1^): ν = 2925 (m), 2862 (w), 1701 (s), 1540 (s), 1488 (s), 1485 (m), 1431 (s), 1375 (m), 1293 (s), 1241 (s), 1205 (s), 1191 (s), 1161 (m), 1147 (m), 1101 (s), 1053 (s), 998 (m), 965 (m), 917 (m), 876 (w), 845 (w), 802 (m), 792 (m), 780 (s), 738 (m). *m*/*z*: 390 (10%), 375 (4%), 359 (13%), 343 (4%), 331 (2%), 237 (21%), 223 (88%), 209 (17%), 195 (25%), 181 (19%), 167 (100%).

*Bis(pyrazol-1-yl)methane-4,4′-dicarboxylic acid* (**1e**) was prepared similarly to compound **1b** by Method B from diester **1f**. Yield 90%, colorless crystals, m.p. 335–337 °C (reprecipitation; dec.). C_9_H_8_N_4_O_4_ (236.19): calcd. C 45.77; H 3.41; N 23.72; found C 45.9; H 3.6; N 23.5. ^1^H-NMR (500 MHz, (CD_3_)_2_SO): δ 6.49 (s, 2 H, CH_2_), 7.87 (s, 2 H, 3-H), 8.53 (s, 2 H, 5-H), 12.52 (s, 2 H, COOH), ppm. ^13^C-NMR (125 MHz, (CD_3_)_2_SO): δ 64.6, 115.9, 134.6, 141.7, 163.4, ppm. FT-IR (cm^−1^): ν = 3108 (m), 1669 (s), 1560 (s), 1419 (m), 1400 (m), 1366 (m), 1345 (m), 1294 (m), 1241 (s), 1133 (m), 1003 (m), 995 (s), 944 (m), 915 (s), 863 (m), 784 (m), 769 (s), 729 (m).

One-pot procedure. A suspension of finely powdered KOH (1.68 g, 30 mmol) and ethyl 4-pyrazole carboxylate (1.40 g, 10 mmol) in 5 mL of DMSO was vigorously stirred for 30 min at 80 °C. After that, the reaction mixture was cooled to room temperature and solution of dibromomethane (0.87 g, 0.35 mL, 5 mmol) in 5 mL of DMSO was added dropwise with stirring and cooling by water over 30 min. Stirring and heating at 80 °C was continued for additional 4 h (TLC control), then 20 mL of water and 1.12 g (20 mmol) of KOH were added to the reaction flask. Heating was continued until complete dissolution of the initially formed precipitate (1 h), after that, additional 80 mL of water were added and the resulting solution was acidified by concentrated HCl. The precipitate was filtered, washed with water and dried. Yield 1.34 g, 92%.

*1,3-Bis(pyrazol-1-yl)propane-4,4′-dicarboxylic acid* (**3e**) was prepared similarly to compound **1b** by Method A (yield 11%) or by Method B from diester **3f**. Yield 76%, colorless crystals, m.p. 225–227 °C (EtOH). C_11_H_12_N_4_O_4_ (264.24): calcd. C 50.00; H 4.58; N 21.20; found C 50,4, H 4,8, N 21,1. ^1^H-NMR (400 MHz, (CD_3_)_2_SO): δ 2.35 (p, 2 H, β-CH_2_), 4.22 (t, 4 H, α-CH_2_), 8.07 (s, 2 H, 5-H), 8.60 (s, 2 H, 3-H), ppm. ^13^C-NMR (100 MHz, (CD_3_)_2_SO): δ 29.9, 49.0, 118.8, 137.0, 141.4, 179.4, ppm. FT-IR (cm^−1^): ν = 3127 (w), 3093 (w), 2948 (w), 1695 (s), 1556 (s), 1473 (w), 1369 (w), 1407 (m), 1356 (m), 1343 (m), 1298 (m), 1224 (s), 1172 (s), 1149 (m), 1083 (m), 1051 (m), 982 (s), 889 (s), 863 (s), 822 (m), 765 (s), 743 (s).

*1,4-Bis(pyrazol-1-yl)butane-4,4′-dicarboxylic acid* (**4e**) was prepared similarly to compound **1b** by Method B from diester **4f**. Yield 86%, colorless crystals, m.p. 300 °C (dec.). C_12_H_14_N_4_O_4_ (278.27): calcd. C 51.80; H 5.07; N 20.13; found C 51.6; H 5.1; N 20.3. ^1^H-NMR (500 MHz, (CD_3_)_2_SO): δ 1.71 (p, 4 H, β-CH_2_), 4.15 (t, 4 H, α-CH_2_), 7.79 (s, 2 H, 5-H), 8.25 (s, 2 H, 3-H), 12.30 (s, 2 H, COOH), ppm. ^13^C-NMR (125 MHz, (CD_3_)_2_SO): δ 26.5, 50.8, 114.6, 133.5, 140.4, 163.8, ppm. FT-IR (cm^−1^): ν = 3122 (w), 3088 (w), 2953 (w), 1685 (s), 1558 (s), 1438 (m), 1404 (m), 1379 (m), 1349 (m), 1304 (m), 1237 (s), 1221 (s), 1188 (s), 1149 (m), 1122 (m), 1018 (s), 996 (m), 986 (s), 947 (m), 896 (m), 867 (m), 805 (s), 775 (s), 748 (s).

*1,5-Bis(pyrazol-1-yl)pentane-4,4′-dicarboxylic acid* (**5e**) was prepared similarly to compound **1b** by Method B from diester **5f**. Yield 67%, colorless powder, m.p. 216–219 °C (reprecipitation; dec.). C_13_H_16_N_4_O_4_ (292.30): calcd. C 53.42; H 5.52; N 19.17; found C 53.6; H 5.7; N 19.0. ^1^H-NMR (500 MHz, (CD_3_)_2_SO): δ 1.15 (p, 2 H, γ-CH_2_), 1.80 (p, 4 H, β-CH_2_), 4.11 (t, 4 H, α-CH_2_), 7.78 (s, 2 H, 5-H), 8.24 (s, 2 H, 3-H), 12.27 (s, 2 H, COOH), ppm. ^13^C-NMR (125 MHz, (CD_3_)_2_SO): δ 22.7, 28.9, 51.2, 114.5, 133.5, 140.3, 163.8, ppm. FT-IR (cm^−1^): ν = 3128 (w), 2974 (w), 1696 (s), 1554 (s), 1472 (w), 1438 (w), 1403 (m), 1376 (w), 1343 (w), 1333 (w), 1298 (w), 1269 (w), 1217 (s), 1098 (m), 1041 (w), 998 (m), 987 (s), 915 (m), 885 (s), 769 (s), 745 (s), 672 (m).

*1,6-Bis(pyrazol-1-yl)hexane-4,4′-dicarboxylic acid* (**6e**) was prepared similarly to compound **1b** by Method B from diester **6f**. Yield 73%, colorless powder, m.p. 252–254 °C (reprecipitation; dec.). C_14_H_18_N_4_O_4_ (306.32): calcd. C 54.89; H 5.92; N 18.29; found C 55.0; H 6.1; N 18.0. ^1^H-NMR (500 MHz, (CD_3_)_2_SO): δ 1.22 (p, 4 H, γ-CH_2_), 1.75 (p, 4 H, β-CH_2_), 4.10 (t, 4 H, α-CH_2_), 7.78 (s, 2 H, 5-H), 8.24 (s, 2 H, 3-H), 12.27 (s, 2 H, COOH), ppm ^13^C-NMR (125 MHz, (CD_3_)_2_SO): δ 25.3, 29.3, 51.4, 114.5, 133.4, 140.3, 163.8, ppm. FT-IR (cm^−1^): ν = 3114 (w), 2940 (w), 2861 (w), 1652 (m), 1549 (w), 1497 (w), 1469 (w), 1438 (w), 1384 (w), 1352 (w), 1258 (w), 1228 (w), 1129 (w), 1053 (w), 995 (w), 982 (w), 931 (w), 860 (w), 778 (m), 736 (w).

*Bis(4-ethoxycarbonylpyrazol-1-yl)methane* (**1f**). A suspension of finely powdered KOH (1.68 g, 30 mmol) and ethyl 4-pyrazole carboxylate (1.40 g, 10 mmol) in 5 mL of DMSO was vigorously stirred for 30 min at 80 °C. After that, the reaction mixture was cooled to room temperature and solution of dibromomethane (0.87 g, 0.35 mL, 5 mmol) in 5 mL of DMSO was added dropwise with stirring and cooling by water over 30 min. Stirring and heating at 80 °C was continued for additional 4 h (TLC control), and the reaction mixture was poured into 100 mL of water, the precipitate was filtered and washed by water. Yield 1.30 g, 89%, colorless crystals, m.p. 144–145 °C (EtOAc/hexane, 1:1). C_13_H_16_N_4_O_4_ (292.30): calcd. C 53.42; H 5.52; N 19.17; found C 53.6; H 5.7; N 18.9. ^1^H-NMR (400 MHz, CDCl_3_): δ 1.34 (t, 6 H, CH_3_), 4.29 (q, 4 H, OCH_2_), 6.31 (s, 2 H, α-CH_2_), 7.96 (s, 2 H, 3-H), 8.18 (s, 2 H, 5-H), ppm. ^13^C-NMR (100 MHz, CDCl_3_): δ 14.3, 60.5, 65.7, 117.0, 133.3, 142.5, 162.3, ppm. FT-IR (cm^−1^): ν = 3149 (w), 3099 (w), 3069 (w), 2987 (w), 1724 (s), 1705 (s), 1558 (s), 1478 (w), 1445 (m), 1412 (w), 1383 (m), 1345 (w), 1290 (m), 1192 (s), 1230 (s), 1161 (m), 1133 (s), 1109 (m), 1029 (s), 1015 (s), 993 (s), 979 (m), 953 (m), 908 (m), 886 (m), 827 (m), 764 (s), 739 (s), 652 (m). *m*/*z*: 292 (23%), 247 (68%), 219 (13%), 153 (100%).

*1,3-Bis(4-ethoxycarbonylpyrazol-1-yl)propane* (**3f**) was prepared similarly to compound **1f**. Yield 80%, colorless crystals, m.p. 92–94 °C (EtOAc/hexane, 1:1). C_15_H_20_N_4_O_4_ (320.35): calcd. C 56.24; H 6.29; N 17.49; found C 56.5; H 6.0; N 17.2. ^1^H-NMR (400 MHz, CDCl_3_): δ 1.36 (t, 6 H, CH_3_), 2.48 (p, 2 H, β-CH_2_), 4.16 (t, 4 H, α-CH_2_), 4.30 (q, 4 H, OCH_2_), 7.94 (s, 2 H, 5-H), 7.95 (s, 2 H, 3-H), ppm. ^13^C-NMR (100 MHz, CDCl_3_): δ 14.4, 30.4, 49.0, 60.3, 115.3, 133.1, 141.3, 162.8, ppm. FT-IR (cm^−1^): ν = 3126 (w), 3091 (w), 2984 (w), 2940 (w), 2875 (w), 1691 (s), 1556 (s), 1472 (w), 1448 (w), 1405 (m), 1373 (s), 1350 (m), 1131 (m), 1286 (m), 1238 (s), 1124 (s), 1218 (s), 1111 (m), 1023 (s), 1084 (s), 976 (s), 874 (s), 830 (m), 769 (s), 708 (w), 653 (m). *m*/*z*: 320 (3%), 291 (1%), 275 (24%), 181 (26%), 167 (100%), 153 (76%), 139 (13%).

*1,4-Bis(4-ethoxycarbonylpyrazol-1-yl)butane* (**4f**) was prepared similarly to compound **1f**. Yield 86%, colorless crystals, m.p. 121 °C (EtOAc/hexane, 1:1). C_16_H_22_N_4_O_4_ (334.38): calcd. C 57.47; H 6.63; N 16.76; found C 57.5; H 6.5; N 16.9. ^1^H-NMR (400 MHz, CDCl_3_): δ 1.27 (t, 6 H, CH_3_), 1.81 (p, 4 H, β-CH_2_), 4.06 (t, 4 H, α-CH_2_), 4.22 (q, 4 H, OCH_2_), 7.78 (s, 2 H, 5-H), 7.83 (s, 2 H, 3-H), ppm. ^13^C-NMR (100 MHz, CDCl_3_): δ 14.4, 27.0, 51.8, 60.2, 115.2, 132.5, 141.2, 162.9, ppm. FT-IR (cm^−1^): ν = 3122 (w), 3089 (w), 2980 (w), 2948 (w), 2869 (w), 1691 (s), 1554 (s), 1464 (m), 1432 (w), 1405 (m), 1394 (m), 1373 (s), 1355 (w), 1338 (w), 1298 (w), 1276 (m), 1223 (s), 1194 (s), 1172 (s), 1113 (s), 1092 (m), 1018 (s), 984 (s), 884 (s), 832 (m), 770 (s), 740 (m), 657 (m). *m*/*z*: 334 (2%), 305 (19%), 289 (50%), 195 (44%), 181 (19%), 167 (59%), 153 (100%), 139 (10%).

*1,5-Bis(4-ethoxycarbonylpyrazol-1-yl)pentane* (**5f**) was prepared similarly to compound **1f**. Yield 95%, colorless crystals, m.p. 82–83 °C (EtOAc/hexane, 1:1). C_17_H_24_N_4_O_4_ (348.40): calcd. C 58.61; H 6.94; N 16.08; found C 58.6; H 6.7; N 16.0. ^1^H-NMR (400 MHz, CDCl_3_): δ 1.21 (p, 2 H, γ-CH_2_, partially overlapped with 1.28), 1.28 (t, 6 H, CH_3_), 1.84 (p, 4 H, β-CH_2_), 4.05 (t, 4 H, α-CH_2_), 4.22 (q, 4 H, OCH_2_), 7.78 (s, 2 H, 5-H), 7.83 (s, 2 H, 3-H), ppm. ^13^C-NMR (100 MHz, CDCl_3_): δ 14.4, 23.4, 29.5, 52.2, 60.2, 115.0, 132.5, 141.0, 163.0, ppm. FT-IR (cm^−1^): ν = 3129 (w), 2970 (w), 2935 (w), 2874 (w), 1699 (s), 1554 (s), 1459 (m), 1441 (m), 1404 (s), 1372 (m), 1352 (m), 1333 (w), 1304 (m), 1263 (w), 1224 (s), 1209 (s), 1190 (s), 1113 (s), 1096 (s), 1021 (s), 980 (s), 877 (m), 826 (w), 762 (s). *m*/*z*: 348 (11%), 319 (24%), 303 (47%), 275 (4%), 209 (14%), 195 (47%), 181 (22%), 167 (33%), 153 (100%), 139 (4%).

*1,6-Bis(4-ethoxycarbonylpyrazol-1-yl)hexane* (**6f**) was prepared similarly to compound **1f**. Yield 73%, colorless crystals, m.p. 85–86 °C (EtOAc/hexane, 1:1). C_18_H_26_N_4_O_4_ (362.43): calcd. C 59.65; H 7.23; N 15.46; found C 59.8; H 7.4; N 15.2. ^1^H-NMR (400 MHz, CDCl_3_): δ 1.24 (p, 4 H, γ-CH_2_, partially overlapped with 1.27), 1.27 (t, 6 H, CH_3_), 1.80 (p, 4 H, β-CH_2_), 4.04 (t, 4 H, α-CH_2_), 4.22 (q, 4 H, OCH_2_), 7.79 (s, 2 H, 5-H), 7.83 (s, 2 H, 3-H), ppm. ^13^C-NMR (100 MHz, CDCl_3_): δ 14.4, 25.9, 29.8, 52.4, 60.2, 115.0, 132.4, 140.9, 163.0, ppm. FT-IR (cm^−1^): ν = 3130 (w), 2983 (w), 2940 (w), 2866 (w), 1693 (s), 1546 (s), 1466 (w), 1442 (w), 1408 (w), 1373 (m), 1352 (w), 1256 (s), 1219 (s), 1201 (s), 1123 (s), 1110 (m), 1052 (m), 1028 (s), 995 (m), 978 (s), 899 (m), 869 (w), 830 (m), 775 (s), 739 (m), 654 (m). *m*/*z*: 362 (10%), 333 (11%), 317 (35%), 289 (4%), 223 (7%), 209 (44%), 195 (29%), 181 (19%), 167 (22%), 153 (100%), 139 (3%).

*Bis(4-ethoxycarbonyl-3,5-dimethylpyrazol-1-yl)methane* (**1g**). A suspension of finely powdered KOH (1.68 g, 30 mmol) and ethyl 3,5-dimethyl-4-pyrazolecarboxylate (1.68 g, 10 mmol) in 5 DMSO was vigorously stirred for 30 min at 80 °C. After that, the reaction mixture was cooled to room temperature and solution of dibromomethane (0.87 g, 0.35 mL, 5 mmol) in 5 mL of DMSO was added dropwise under stirring and cooling by water over 30 min. Stirring and heating at 80 °C was continued for additional 4 h (TLC control), and the reaction mixture was poured into 100 mL of water, the precipitate was filtered and washed by water. Yield 1.51 g, 87%, colorless crystals, m.p. 152–153 °C (EtOAc/hexane, 1:2). C_17_H_24_N_4_O_4_ (348.18): calcd. C 58.61; H 6.94; N 16.08; found C 58.4; H 7.1; N 16.2. ^1^H-NMR (400 MHz, CDCl_3_): δ 1.35 (t, 6 H, OCH_2_CH_3_), 2.39 (s, 6 H, 3-CH_3_), 2.75 (s, 6 H, 5-CH_3_), 4.29 (q, 4 H, OCH_2_), 6.12 (s, 2 H, α-CH_2_), ppm. ^13^C-NMR (100 MHz, CDCl_3_): δ 11.4, 14.3, 14.4, 59.8, 59.9, 110.9, 145.8, 151.4, 164.2, ppm. FT-IR (cm^−1^): ν = 2990 (w), 2937 (w), 1701 (s), 1557 (s), 1476 (m), 1426 (m), 1394 (m), 1352 (m), 1309 (s), 1299 (m), 1241 (s), 1212 (m), 1166 (s), 1121 (m), 1101 (s), 1042 (m), 1002 (m), 853 (m), 815 (m), 782 (s), 735 (m), 691 (s). *m*/*z*: 348 (50%), 333 (5%), 303 (33%), 275 (2%), 181 (100%).

*1,3-Bis(4-ethoxycarbonyl-3,5-dimethylpyrazol-1-yl)propane* (**3g**) was prepared similarly to compound **1g**. Yield 92%, colorless crystals, m.p. 112–113 °C (EtOAc/hexane, 1:1). C_19_H_28_N_4_O_4_ (376.46): calcd. C 60.62; H 7.50; N 14.88; found C 60.3; H 7.7; N 14.9.^1^H-NMR (400 MHz, CDCl_3_): δ 1.36 (t, 6 H, OCH_2_CH_3_), 2.38 (p, 2 H, β-CH_2_, partially overlapped with 2.41), 2.41 (s, 6 H, 3-CH_3_), 2.46 (s, 6 H, 5-CH_3_), 4.04 (t, 4 H, α-CH_2_), 4.29 (q, 4 H, OCH_2_), ppm. ^13^C-NMR (100 MHz, CDCl_3_): δ 11.0, 14.3, 14.4, 29.3, 45.5, 59.6, 109.7, 144.0, 150.6, 164.4, ppm. FT-IR (cm^−1^): ν = 2980 (w), 2929 (w), 1712 (m), 1681 (m), 1545 (m), 1489 (m), 1425 (m), 1385 (m), 1370 (m), 1323 (w), 1293 (s), 1249 (w), 1231 (m), 1200 (w), 1138 (s), 1120 (m), 1100 (s), 1043 (m), 1013 (m), 1000 (m), 984 (w), 853 (w), 815 (w), 782 (s), 716 (w). *m*/*z*: 376 (11%), 331 (10%), 209 (18%), 195 (100%), 181 (51%), 167 (6%).

*1,4-Bis(4-ethoxycarbonyl-3,5-dimethylpyrazol-1-yl)butane* (**4g**) was prepared similarly to compound **1g**. Yield 86%, colorless crystals, m.p. 122–123 °C (EtOAc/hexane, 1:1). C_20_H_30_N_4_O_4_ (390.48): calcd. C 61.52; H 7.74; N 14.35; found C 61.7; H 8.0; 14.4. ^1^H-NMR (400 MHz, CDCl_3_): δ 1.36 (t, 6 H, OCH_2_CH_3_), 1.82 (p, 4 H, β-CH_2_), 2.42 (s, 6 H, 3-CH_3_), 2.49 (s, 6 H, 5-CH_3_), 4.03 (t, 4 H, α-CH_2_), 4.29 (q, 4 H, OCH_2_), ppm. ^13^C-NMR (100 MHz, CDCl_3_): δ 11.1, 14.3, 14.4, 26.9, 48.0, 59.6, 109.7, 143.7, 150.4, 164.4, ppm. FT-IR (cm^−1^): ν = 2985 (w), 2933 (w), 2875 (w), 1695 (s), 1547 (s), 1491 (m), 1472 (m), 1436 (m), 1400 (m), 1383 (m), 1364 (m), 1298 (s), 1248 (s), 1188 (s), 1122 (m), 1102 (s), 1076 (m), 1044 (m), 1015 (m), 993 (m), 910 (w), 888 (w), 849 (m), 813 (w), 784 (s), 742 (w), 711 (m). *m*/*z*: 390 (36%), 375 (12%), 361 (12%), 345 (29%), 329 (1%), 317 (4%), 223 (25%), 209 (29%), 195 (68%), 181 (100%).

*1,5-Bis(4-ethoxycarbonyl-3,5-dimethylpyrazol-1-yl)pentane* (**5g**) was prepared similarly to compound **1g**. Yield 79%, colorless crystals, m.p. 100–101 °C (EtOAc/hexane, 1:1). C_21_H_32_N_4_O_4_ (404.51): calcd. C 62.35; H 7.97; N 13.85; found C 62.0; H 8.1; N 13.9. ^1^H-NMR (400 MHz, CDCl_3_): δ 1.32 (p, 2 H, γ-CH_2_, partially overlapped with 1.36), 1.36 (t, 6 H, OCH_2_CH_3_), 1.81 (p, 4 H, β-CH_2_), 2.42 (s, 6 H, 3-CH_3_), 2.49 (s, 6 H, 5-CH_3_), 3.99 (t, 4 H, α-CH_2_), 4.29 (q, 4 H, OCH_2_), ppm. ^13^C-NMR (100 MHz, CDCl_3_): δ 11.2, 14.3, 14.4, 23.7, 29.5, 48.5, 59.6, 109.5, 143.5, 150.3, 164.5, ppm. FT-IR (cm^−1^): ν = 2981 (w), 2930 (w), 2866 (w), 1686 (s), 1548 (s), 1496 (m), 1473 (m), 1436 (m), 1399 (w), 1371 (m), 1296 (s), 1281 (m), 1246 (s), 1219 (m), 1190 (s), 1166 (m), 1122 (s), 1101 (s), 1042 (m), 997 (m), 889 (w), 852 (m), 812 (w), 783 (s), 746 (w), 718 (w), 698 (w). *m*/*z*: 404 (29%), 389 (8%), 375 (10%), 359 (30%), 343 (6%), 331 (4%), 237 (28%), 223 (47%), 209 (35%), 195 (47%), 181 (100%).

*1,6-Bis(4-ethoxycarbonyl-3,5-dimethylpyrazol-1-yl)hexane* (**6g**) was prepared similarly to compound **1g**. Yield 81%, colorless crystals, m.p. 107–108 °C (EtOAc/hexane, 1:1). C_22_H_34_N_4_O_4_ (418.54): calcd. C 63.13; H 8.19; N 13.39; found C 63.2; H 8.3; N 13.4. ^1^H-NMR (400 MHz, CDCl_3_): δ 1.33 (p, 4 H, γ-CH_2_, partially overlapped with 1.36), 1.36 (t, 6 H, OCH_2_CH_3_), 1.79 (p, 4 H, β-CH_2_), 2.42 (s, 6 H, 3-CH_3_), 2.49 (s, 6 H, 5-CH_3_), 3.98 (t, 4 H, α-CH_2_), 4.29 (q, 4 H, OCH_2_), ppm. ^13^C-NMR (100 MHz, CDCl_3_): δ 11.2, 14.3, 14.4, 26.2, 29.8, 48.6, 59.6, 109.4, 143.5, 150.2, 164.6, ppm. FT-IR (cm^−1^): ν = 2983 (w), 2929 (w), 2863 (w), 1686 (s), 1548 (s), 1481 (m), 1469 (m), 1436 (m), 1374 (m), 1361 (m), 1320 (m), 1311 (m), 1291 (s), 1238 (s), 1185 (s), 1121 (s), 1102 (s), 1041 (m), 1018 (m), 997 (m), 851 (m), 819 (w), 783 (s), 744 (w), 716 (m). *m*/*z*: 418 (21%), 403 (2%), 389 (6%), 373 (29%), 357 (8%), 345 (4%), 251 (25%), 237 (100%), 223 (22%), 209 (26%), 195 (20%), 181 (87%).

*Bis(pyrazol-1-yl)methane-4-carboxylic acid* (**7**) was prepared following the procedure for compound 1b (Method A). Yield 10%, colorless crystals, m.p. 205–207 °C (EtOH). C_8_H_8_N_4_O_2_ (192.18): calcd. C 50.00; H 4.20; N 29.15; found C 50.4; H 4.5; N 28.8. ^1^H-NMR (400 MHz, (CD_3_)_2_SO): δ 6.32 (t, 1 H, 4-H-Pz), 6.50 (s, 2 H, CH_2_), 7.53 (d, 1 H, 3-H-Pz), 8.01 (d, 1 H, 5-H-Pz), 8.08 (s, 1 H, 3-H-PzCOOH), 8.12 (s, 1 H, 5-H-PzCOOH), ppm. ^13^C-NMR (100 MHz, (CD_3_)_2_SO): δ 64.5, 106.7, 119.7, 131.4, 136.8, 140.9, 142.2, 179.6, ppm. FT-IR (cm^−1^): ν = 3430 (br., COOH), 1682 (C=O), 1544 (ν_Pz_), 1002 (Pz breathing).

## 4. Conclusions

In summary, approaches to the synthesis of 4,4′-dicarboxy-substituted bis(pyrazol-1-yl)alkanes were evaluated. It was found that direct carboxylation by oxalyl chloride is feasible only for the preparation of bis(3,5-dimethylpyrazol-1-yl)methane derivates due to electron-donating methyl groups and short methylene linker. Longer linkers lead to significantly lower product yields, which is probably due to the instability of the intermediate acid chlorides that are formed in the reaction with oxalyl chloride. A more universal method is based on the reaction of ethyl 4-pyrazolecarboxylates with dibromoalkanes in a superbasic medium and is applicable for the preparation of bis(4-carboxypyrazol-1-yl)alkanes with both short and long linkers. The obtained dicarboxylic acids are interesting as potential building blocks for metal-organic frameworks.

## Data Availability

The data presented in this study are available on request from the corresponding author.

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
