# Peer review of "Approaches to the Synthesis of Dicarboxylic Derivatives of Bis(pyrazol-1-yl)alkanes"

_molecules, 2021, doi:10.3390/molecules26020413_

Round 1
Reviewer 1 Report
The manuscript “Approaches to the synthesis of dicarboxylic derivatives of bis(pyrazol-1-yl)alkanes” describes the preparation of relevant pyrazole derivatives useful as synthetic intermediates or as compounds with biological properties. The work is well organized, well written and the experimental data are consistent. In the Introduction, a brief review of methods to obtain carboxylic derivatives of pyrazoles is advisable. It is recommendable to cite references such as Mini-Reviews in Organic Chemistry, 2010, 7, 314. Furthermore, some typing mistakes should be sought and corrected such at line 10: “If was found…” should be “It was found…”. In summary, the manuscript is perfectly suitable for publication in Molecules after considering the above-mentioned issues.
Author Response
- In the Introduction, a brief review of methods to obtain carboxylic derivatives of pyrazoles is advisable. It is recommendable to cite references such as Mini-Reviews in Organic Chemistry, 2010, 7, 314.
Methods of synthesis of pyrazole-carboxylic acids are briefly reviewed in lines 47-50, refs. 29-33. The suggested and very useful review was cited as ref. 33.
- Furthermore, some typing mistakes should be sought and corrected such at line 10: “If was found…” should be “It was found…”. - corrected
Reviewer 2 Report
Manuscript ID: molecules-1073860
Type of manuscript: Article
Title: Approaches to the synthesis of dicarboxylic derivatives of bis(pyrazol-1-yl)alkanes
Authors: Nikita P. Burlutskiy, Andrei S. Potapov
This manuscript describes the synthesis of bis(pyrazol-1-yl)alkanes decorated with a COOH functional group on each pyrazol ring. The synthetic methodology has been studied and well optimized taking into account the effects of the different groups (H, Me) on the heterocycles rings, the length of the alkyl chain which connects the two pyrazols and the hydrolysis conditions. Moreover, the different behavior of some of the substrates has been investigated using DEFT calculations of charge distribution.
The manuscript is well organized, the obtained results are interesting, a detailed study on the optimization of the experimental conditions and substrates scope is reported and discussed. As a consequence I find this manuscript suitable for publication to Molecules as it is.
Author Response
Thank you for your high evaluation of our work.